# New Sustainable Process for Hesperidin Isolation and Anti-Ageing Effects of Hesperidin Nanocrystals

**DOI:** 10.3390/molecules25194534

**Published:** 2020-10-03

**Authors:** Danijela Stanisic, Leticia H. B. Liu, Roney V. dos Santos, Amanda F. Costa, Nelson Durán, Ljubica Tasic

**Affiliations:** 1Biological Chemistry Laboratory, Organic Chemistry Department, Institute of Chemistry, University of Campinas (UNICAMP), Campinas 13083-970, SP, Brazil; dacici.stanisic@gmail.com (D.S.); leticia.bacellar@gmail.com (L.H.B.L.); torroney@gmail.com (R.V.d.S.); amanda.fc92@gmail.com (A.F.C.); 2Laboratory of Urogenital Carcinogenesis and Immunotherapy, Department of Structural and Functional Biology, University of Campinas, Campinas 13083-862, SP, Brazil; nelsonduran1942@gmail.com; 3Nanomedicine Research Unit (Nanomed), Federal University of ABC (UFABC), Santo André 09210-580, SP, Brazil

**Keywords:** hesperidin, new-clean process extraction, nanocrystals, antioxidant, anti-ageing

## Abstract

Hesperidin, a secondary orange (*Citrus sinensis*) metabolite, was extracted from orange bagasse. No organic solvents or additional energy consumption were used in the clean and sustainable process. Hesperidin purity was approximately 98% and had a yield of 1%. Hesperidin is a known supplement due to antioxidant, chelating, and anti-ageing properties. Herein, hesperidin application to eliminate dark eye circles, which are sensitive and thin skin regions, was studied. In addition, the proposed method for its aqueous extraction was especially important for human consumption. Further, the most effective methods for hesperidin nanonization were explored, after which the nanoemulsions were incorporated into a cream formulation that was formulated for a tropical climate. Silky cream formulations (oil in water) were tested in vitro on artificial 3D skin from cultured cells extracted from skin residues after plastic surgery. The proposed in vitro assay avoided tests of the different formulations in human volunteers and animals. It was shown that one of the nanonized hesperidin formulations was the most skin-friendly and might be used in cosmetics.

## 1. Introduction

Hesperidin (HSD) is a secondary plant metabolite and one of the principal bioflavonoids of citrus fruits. This flavanone and its aglycone form, hesperetin (HST), are present in relatively high quantities (1–2%), especially in sweet immature oranges (*Citrus sinensis*) [1,2]. HSD is known for valuable bioactivity and can act as an antioxidant [3,4,5,6,7,8,9,10], anti-inflammatory [3,9,10], hypolipidemic [8], vasoprotective [3,4,8,11], and anticancerogenic [3,9] agent. The great majority of flavanones are glycosylated mainly with rutinose and neohesperidose [1,12]. HSD (Figure 1) is proposed, after different in vitro studies, as a plant defense molecule and a potent antioxidant [7,13]. It is water soluble just at high pH, dimethyl sulfoxide, and pyridine, giving yellow and clear solutions. HSD is partially soluble in methanol and glacial acetic acid, and almost insoluble in acetone, chloroform, and benzene [12,14].

### 1.1. Antioxidant Activity of Hesperidin

Bioflavonoids have been known for a long time because of their noticeable antioxidant potential that mitigates the harmful impact of free radical and reactive oxygen species. Hydroxyl, alkoxyl, peroxyl, and peroxyl nitrite radicals induce ageing, tissue damage, and contribute to many disease developments like cancer, hypertension, atherosclerosis, amyloidosis, and senile dementia. Nevertheless, the cited free radicals are produced in everyday life and neutralized as well by enzymes [1]; with ageing and increased radical production, their nocive effects are enlarged and become far more potent. The radical scavenging property inherent to natural bioflavonoids is related to their structure and in HSD, hydroxyl groups at positions 3′- and 5- demonstrate mild antioxidant activities [15,16]. Hesperetin (HST) has an additional hydroxyl group (7-), thus showing a stronger antioxidant activity compared to HSD. Disaccharide in the HSD structure affects its capacity for electronic delocalization and decreases its antioxidant activity [6]. For example, HSD nano-loaded lipid carrier showed antioxidant effects in concentrations of 45 µM in 2,2-diphenyl-1-picrylhydrazyl (DPPH) assay, and without any toxic effects on several cell lines [17].

### 1.2. Hesperidin and Hesperetin Use in Cosmetics as Anti-Ageing Active Components

Hesperidin and hesperetin are the foci of intensive research for topical application. For example, sulfonated, acetylated, or phosphorylated HSD derivatives are potent inhibitors of hyaluronidase. Moreover, HSD can also act on superoxide in electron transfer as well as proton transfer in vivo. HSD might act as a topical UV-protective agent, by protecting phosphatidylcholine liposomes from UV irradiation-induced peroxidation. The neohesperidin, for example, demonstrates the capacity to extend yeast’s chronical lifespan, individually or in synergism with the HST, for 10 different aging factors, such as scavenging ROS effects, regulation of stress-related enzymes, and maintaining pH cellular value, favorable for life extension of yeast cells [18]. Hesperidin was also proved as a potent anti-photoageing factor, through regulation of metalloproteinases MMP-9 via mitogen activation protein kinase (MAPK) signaling pathways. In the same study, Lee and colleagues [19] approved the positive effect of the hesperidin on wrinkle depth on a mouse dorsal skin model, and reduction in UVB-induced hydration changes and trans-epidermal water loss [20]. Different studies demonstrate accelerated cutaneous diabetic or venous wounds healing and ameliorate skin epidermal barrier function already after 7 days of HSD application [20]. On the other side, HST was found to penetrate through the *stratum corneum* [5] and assays conducted in vitro showed that the presence of lecithin and d-limonene in formulations may aid towards faster hesperetin penetration into the skin [21]. Additionally, in vitro studies of the *stratum corneum* have demonstrated that flavonoids show differences in penetration capacities, which depend in a large part on the ingredients/vehicles present in the formulation. Therefore, the penetration rate of flavonoids, such as catechin, rutin, quercetin, and others, is influenced by moisturizing ingredients (glycerol, glycols, polyglycols, ethoxylated methyl glucoside, and urea) and by the type of cosmetic formulation (hydrogel, emulsion, microemulsion, and micellar system) [6]. For example, the water in oil microemulsion formulations significantly enhances quercetin skin penetration up to 12 h after application [22,23,24]. HSD-loaded nanostructured lipid vehicles show burst release in the beginning and further sustained bioflavonoid release from the lipid nanocarrier [17]. Despite their interesting and beneficial skin effects, bioflavonoids are very demanding for making effective formulations. In addition, their insolubility in water complicates their use in cosmetic products and influences greatly on their extraction from fruits and plants that is usually performed by the use of organic solvents.

Many different protocols are described and used for the extraction of hesperidin from various starting materials, including maceration [12,25,26,27], Soxhlet extraction [26,28], assisted extraction with ultrasound [25], high hydrostatic pressure and microwave assisted extraction [29,30], enzymatic process [14], and supercritical fluids extraction [31]. All the aforementioned extraction methods require the use of organic solvents, energy, and/or high-pressure, which makes those processes unfavorable for an ecologically clean way of treating orange peel. Herein, a new and green method, without using organic solvents and with low energy consumption, was implemented for the extraction of hesperidin from fresh orange bagasse. Hesperidin extracted in this manner is safe to use and biocompatible for topical application.

## 2. Results

### 2.1. Yield of Hesperidin Extracted from Humid Orange Peel—Green Method

The obtained hesperidin was pale yellow amorphous powder (Appendix A). The yield of HSD extracted in this manner was ~1.2% calculated per dry orange peel bagasse. The same method was repeated varying calcium(II) chloride solution concentrations—5% and 10%—but HSD yields were significantly lower when compared to one obtained with 7.5% calcium(II) chloride solution. In addition, the color of the solution with 7.5% calcium(II) chloride was much darker than obtained with the other two solutions (Appendix A). In the case of 5%, there are not enough calcium(II) ions to react the pectin and disrupt interaction of HSD with the macromolecule, and in the case of 10% of the solution, calcium(II) interactions with HSD are due to their ability to chelate metal ions.

### 2.2. Spectral Data and Characterization of Obtained HSD

The chemical structure and purity of the obtained HSD were analyzed applying High-field Nuclear Magnetic Resonance (NMR) (Appendix A), Fourier Transform-Infrared (FT-IR) spectroscopy and Ultra High-Performance Liquid Chromatography (UPLC, Appendix A) techniques. All HSD NMR peak assignments of proton and carbon signals were in accordance with the literature [32]. From the ^1^H NMR spectrum of the obtained HSD, typical flavanone structure signals could be assigned at δ 12.02 (1H, s, 5-OH) and δ 9.09 (1H), which originate from two hydroxyl groups attached to an aromatic ring, further δ 5.51 (1H, dd, *J* = 12.1, 3.3 Hz, H-2), from protons of the 1,3,4-trisubstitued ring at δ 6.91 (3H, m), protons of rhamnose and glucose at δ 4.98 (1H, d, *J* = 7.3 Hz) and 4.53 (H, m), one methoxy group peak at δ 3.80 (3H, s), and one methyl group peak at δ 1.09 (4H, d, *J* = 6.2 Hz). The acquired Heteronuclear Single Quantum Coherence (HSQC) spectra (Appendix A) and Heteronuclear Multiple Bond Coherence (HMBC) spectra (Appendix A) of the extracted HSD were used to confirm its structure. HSD was monitored by UV maximum at 284 nm, caused by the conjugation of the keto group and the other oxygen atoms with the aromatic ring systems. HSD purity was determined utilizing Ultra High-Performance Liquid chromatography (UHPLC) with the reverse stationary phase (C18, length 10 cm, 5 μm). HSD analytical standard (Sigma-Aldrich, 97% of purity) was used for calibration curve construction from the injection of aliquots of 1, 2, 4, 6, 8, and 10 μL of standard HSD solution in methanol (concentration 100 ppm). Calibration curves and the analysis of the sample from the extraction were performed in triplicate. A solution of 100 ppm of HSD in methanol (HPLC grade of purity) was analyzed in UHPLC as to obtain the area of the peak and to determine the relative concentration of HSD, comparing to the Sigma-Aldrich standard of HSD with grade purity > 97%. The peak with the retention time of 6.877 min was assigned to HSD (Appendix A) as the same procedure was done with the standard for HSD (>97%, Sigma-Aldrich). The determined purity was 97.2%, m/m, when compared to the Sigma-Aldrich analytical standard (> 97%). Therefore, the applied extraction process was adequate and a high HSD yield was achieved, while purity of the extracted HSD was high. It was observed that low impurities coming from naringin were present (Appendix A).

### 2.3. Chelation Properties of Hesperidin

Hesperidin and copper(II) acetate were dissolved in methanol and mixed to obtain hesperidin-copper complex. Likewise, with other compounds with a similar structure, HSD possesses chelation potential when it is mixed with metal(II) ions [33]. Figure 2 presents the FT-IR spectra of hesperidin, copper(II) acetate, and hesperidin–copper complex.

The FT-IR spectra of HSD, copper(II) acetate, and the HSD–Cu(II) complex showed HSD band shift upon complexation with a metal ion [33]. For example, the vibration stretching for -C=O in HSD at 1644 cm^−1^ was shifted to 1515 cm^−1^ in the HSD–Cu(II) complex, due to the coordination with the metal ion. The absorption bands in HSD for C-O at 1297, 1275, 1242, 1206, 1184, 1154, 1132, 1095, 1054, 1037, and 1009 cm^−1^ were suppressed, probably because of the Cu(II) bonding with -OH groups. Therefore, there is a strong indication that hesperidin formed a complex with Cu(II) ions.

### 2.4. HSD Interacts with Collagenase

Collagenase enzymes are a group of metalloproteinases responsible for the degradation of collagen and may cause an ageing effect in skin when present in higher levels in extracellular matrix. Figure 3 presents suppression of fluorescence quenching—observed for collagenase (*Clostridium histolyticum*, Sigma-Aldrich) by the addition of hesperidin in different concentrations (DMSO solution). Hesperidin, as other flavonoids, can reduce the intensity of tryptophan or other fluorophores of enzyme fluorescence emissions. Hesperidin chelates zinc(II) ion, which is in the catalytic site of the enzyme, and changes the conformation of the protein. There are already various studies in which it has been proven that flavonoids have an inhibitory effect on metalloproteinase (collagenase, elastases, and hyaluronidases) by chelating their metal ions. Additionally, the temperature of the experiment was varied as to evaluate and calculate the thermodynamic values of enthalpy (ΔH) and entropy (ΔS) of collagenase–hesperidin interaction.

The mechanism of fluorescence suppression can be explained through collision quenching between the excited enzyme and suppressor (*Q*), defined by bimolecular quenching constant *Kq* according to Stern–Volmer [34]. The reduction in the intensity of the fluorescence emission (*F*) in relation to the intensity observed in the absence of suppressor (*F*_0_) is given by Equation (1):(1)$$(F_{0}/F)=1+K_{q}\tau_{0}\left[Q\right]=1+K_{S}\left[Q\right]$$

[*Q*] is a concentration of HSD, *τ*_0_ is 10^−8^ s, and *Kq* is the association constant, 0.29 × 10^−3^ mol^−1^ L, 0.32 × 10^−3^ mol^−1^ L, or 0.15 × 10^−3^ mol^−1^ L obtained for collagenase–HSD at 25 °C, 30 °C, and 37 °C, respectively. The active site number was estimated at n = 1.2 (1.26, 1.24, and 1.19 for collagenase–HSD interaction at 25 °C, 30 °C, and 37 °C, respectively). It was possible to calculate Gibbs free energy, entropy, and enthalpy, ref [35] for collagenase–HSD at 25 °C, 30 °C, and 37 °C, as shown in Table 1.

According to the data (Table 1) and as well to Ross and Subramanian [35], the interaction between collagenase and HSD is a partial immobilization. This interaction occurs when protein and ligand are leaving the state from which the two are separated and when they are hydrophobically associated. Further, this type of association contributes to the decrease in the ΔS and ΔG values.

### 2.5. Characterization of HSD Nanocrystals

The obtained HSD was pre-milled and further nanonized on a homogenizer or ultrasound of high potency. The effectiveness of pre-milling was clearly revealed by the reduction in particle size distribution observed after applying different analytical methodologies for HSD nanonization. The average size, zeta potential, and polydispersity index of the HSD nanoparticles are shown in Table 2.

The results were obtained after 6 months of preparation of the emulsions. The sizes of the HSD nanoparticles, after passing through the homogenizer and ultrasound, reduced from 600 to 150–350 nm, depending on the type, concentration of the polymer, and technique used for preparation. Zeta potentials were very variable, and it was not possible to obtain the desired stability for all nanoemulsions (for example, III, Table 2) with the best zeta potential being around −30 ± 6 mV. Part of the particles had sizes up to 400 nm as measured using the Dynamic Light Scattering (DLS) and will not affect their permeation via *stratum corneum*.

Scanning and transmission electron micrographs obtained for the formulations I, II, and IV–VIII are presented in Figure 4 and show particles whose sizes are in accordance with the sizes measured by (DLS) and NanoTracking Analysis (NTA) techniques. Nanoparticles of formulation I were around 200 nm in size, as shown in Table 2 and Figure 4. The formulation II contained bigger particles, around 1 µm, probably because of sodium-carboxyl methyl cellulose as surfactant (image not shown). Formulation IV (Figure 4) had particles with sizes of 250 nm (DLS and NTA), probably due to the aggregation of poloxamer around the hesperidin particles, whereas NTA experiments showed the presence of smaller particles of 50–100 nm.

Formulations V and VI were prepared utilizing the technique NANOEDGE^like^, where hesperidin was dissolved in small amounts of propylene glycol and glycerol, respectively. SEM experiments were not adequate to characterize the VI formulation and it was necessary to apply atomic force microscopy (AFM) to perform such characterization. The particles were in the size range of 50–200 nm, and in the image of formulation VI, nanofibers of nanocellulose could be seen. Formulations V and VI showed above average stability for 6 months without any precipitation of the nanoparticles. Their stability was achieved mostly because of the significant amount of nanocellulose used for their production, while good dispersibility of hesperidin was achieved because of propylene glycol and glycerol used in their formulation. As the formulations were prepared in aqueous solutions, there was no observed HSD binding with CNF, which was also especially important from the point of view of skin permeation. The stability of the prepared nanoemulsions (I–VIII) was very good (up to 6 months), which despite showing a small percentage of agglomerates, returned to the initial state by a process involving the use of an ultrasound bath (5 min).

### 2.6. Stability of HSD Cream Formulations

The stability tests for HSD incorporated in silky cream formulations, kept in plastic boxes made of polypropylene at 4 °C and at room temperature, were carried out 72 h, 30 days, and 12 months after manufacture. Organoleptic tests included appearance, color, brightness, consistency, and homogeneity of the creams, evaluated by visual observation. Applied and sensory characteristics of samples, such as spreadability, absorption, stickiness, and fat film on the skin were assessed after applying the cream on the skin. Shortly after preparation, creams were homogenous, bright, and easily spreadable on the skin. During the following 12 months, their consistency did not change, and the cream formulations maintained odor and color as well. Measurement of the pH value was carried out potentiometrically, by direct immersion of the glass electrode into the tested samples of creams at room temperature. Values of the pH were in the range from 5 to 6, as is recommended by pharmacopoeia.

### 2.7. Evaluation of Toxicity of HSD Nanoemulsions and Cream

Full thickness skin (FTS) models for the toxicity (corrosion) test of hesperidin cream formulations were synthesized in the Laboratory of Biology of the Skin, Faculty of Pharmaceutical Sciences, University of Sao Paulo. As the experiment is quite complicated and expensive, just some of the HSD cream formulations were tested.

Figure 5 presents photomicrographs of histological analysis—eosin stain (samples in paraffin) of the FTS skin model after the cream application assay (24 h). This model was synthesized from the fibroblast and keratinocytes isolated from normal human skin cells-donated foreskin samples. The model was composed from two main skin layers-epidermis and dermis-and used for the evaluation of the toxicity of cosmetic and other preparations. Epidermis was composed from five layers: stratum corneum, stratum lucidum, stratum granulosum, stratum spinosum, and stratum basale. The last one was formed from the undifferentiated cells, with a full metabolic activity, which was lost in later proliferation of the cells and formation of the corneocytes. It is important to mention that macrophage and Langerhans cells were not present, which are the first defensive line of the skin [36].

In the negative control sample (Figure 5a) photomicrograph, epidermis and dermis layers were preserved and there was no presence of vacuolization in the stratum basale. After application of the A1 cream (without HSD-nanoemulsion), the basal layer was preserved, with remaining proliferation despite some vacuoles’ presence (Figure 5b). The stratum corneum of this sample was preserved and thicker, and the cream demonstrated good moisturizing effects. HSD in 10% DMSO showed corrosive and cytotoxic effects on the FTS skin model, and led to complete degradation of the basal layer (Figure 5b). Further, the formulation with even a small percentage of DMSO was avoided. Application of nanoemulsion VI showed an irregular basal layer and elevated stratum corneum; therefore, a slight corrosive effect for this formulation was observed (Figure 5d). Nanoemulsion VII demonstrated less cytotoxic effects, inclusively in the basal membrane, compared to the A2 cream. The corrosion test with the A2 and A3 creams (data not shown) showed moderate hydration effects on the FTS skin model, probably due to the hygroscopic effect of some of their ingredients. As this skin model is a living organ, some of the ingredients of the nanoemulsion or the surfactant alone could have led to lower vitality of the cells and prevented their further proliferation. The A2 cream contained only a Pluronic-F127 as a hygroscopic compound, and therefore, the viability of the cells was much higher than in the case of the A3 cream. Nanoemulsions VI and VII demonstrated corrosive effects on the skin model (data not shown) due to the dilution of the buffer solution as well as the inadequate environment for the development and proliferation of cells. It is important to mention that the generated FTS skin model does not contain macrophages or Langerhans cells, which are the first defense of the skin tissue’s immunity, which makes it much more sensitive than normal skin tissue.

## 3. Discussion

A new, water-based, eco-friendly extraction was performed to yield hesperidin (HSD, 1.2%) with approximate purity of 98%. This new method explored the use of calcium(II) and changes in the pH as to isolate HSD from orange peel. The use of calcium(II) enabled the complexation of pectin within the bagasse and release of HSD, which dissolved at high pH values. NaOH solution at pH ~11.5 deprotonated phenol HSD groups and promoted its better solubility. The next step counted on HSD protonation (HCl), which caused HSD precipitation on pH ~4.2. Pale yellow powder was obtained and characterized using different spectroscopic techniques. HSD is usually extracted with organic solvents [30,32] or by supercritical fluid extraction [31]. The organic solvent extraction for hesperidin extraction was performed by Cypriano and colleagues [37] in a 1.2% yield, following Ikan’s method [12]. The method suggested in this work is energy- and organic solvents-free, and calcium(II) and pH-triggered. The use of calcium(II) facilitated complexation of pectin polysaccharide and at the same time, liberation of hesperidin from orange bagasse. The suggested process is longer but facilitates the extraction of hesperidin with greater purity compared to the Sigma-Aldrich analytical standard.

Observing the structure of HSD (Figure 1), it is evident that its B-ring is electron richer than the A-ring due to the contribution of the ortho methoxy group. The first site of oxidation in HSD is the 3′-phenoxyl group in a one-electron reduction [8,38]. The pKa of HSD -OH groups is 8.9 and 11.2, respectively. However, the pKa of the hydroxyl radical in the 3′ position drops to a value around 4–5. Due to this, HSD acts as antioxidant in the pH range of 7 to 10. Therefore, the mechanism of HSD oxidation involves the transfer of one electron and one proton. The reduction potential of the phenolate radical is strongly affected by substituents in the B-ring [8]. HSD has an electron-donating group (methoxy) attached to the B-ring, which elevates its reduction potential (E = 0.72 V) when compared to other flavonoids with catechol groups in the B-ring (E value from 0.5 to 0.7 V). Nevertheless, its reduction potential is still smaller than those from alkyl peroxyl radicals (E = 1.05 V) and superoxide radicals (E = 0.94 V), which confirms that HSD can scavenge and inactivate these harmful radical species in the organism [8,39,40,41,42,43,44,45,46,47,48,49,50,51]. HSD high chemical reactivity can prevent injuries caused by free radicals through direct interaction with radical oxygen species (ROS), or indirectly by inducing antioxidant enzymes activation, chelating of metal ions, reduction in α-tocopherol radicals, inhibition of oxidases, reduction in NO oxidative [4,38] stress, and increases in antioxidant effects of low molecular antioxidants [9,39].

Many other polyphenols, such as phenolic acid, flavonoids, coumarins, stilbenes, and lignans, are the most important polyphenols used in traditional and modern cosmetic and dermatologic products [3,52,53]. The antioxidant, anti-ageing, anti-inflammatory, antimicrobial, and anticancer properties of flavonoids are frequently deployed in skincare products. Their antiradical functions, as well as the ability to inhibit some enzymes, are some of the most important effects. Flavonoids have an influence on skin microcirculation and can be used as ingredients in creams for vascular, oily, and/or atypical skin [3]. Dermal bioavailability and antioxidant activity [54,55] of glycosides is beneath when compared to their aglycon forms, due to their low permeability [3]. One of the interesting physical-chemical properties of bioflavonoids is their ability to absorb ultraviolet radiation due to the presence of the conjugated double bonds in their structure. They may absorb UV radiations in the range of 550–300 and 285–240 nm, resulting from phenolic electron-donating substituents, as well as inter- and intramolecular hydrogen bonds and steric effects. The polyphenol concentrations used in cosmetics cannot replace conventional UV filters. Moreover, they can act as co-adjuvants in erythema and skin burns reduction caused by exposures to UVB radiation and IR rays of sunlight [3]. The anti-ageing properties of flavonoids are associated with their effect on the modulation of matrix metalloproteinases activities, which are dependent on the Zn^2+^ ion and involved in connective tissue remodeling. Equally, by sequestering metal ions or the effects on the expression of endogenous protein tissue inhibitors of metalloproteinase (TIMPs) [9], flavonoids can act protectively. Flavonoids’ skin whitening activity relates to their modulation of tyrosinase activity, through their chelating of calcium(II), in the formation of L-dopaquinone, which is a part of the melanogenesis process [56]. This ability is common to quercetin, cyanidin, and kaempferol, while their glycoside forms have not shown this capability [3]. Hyaluronic acid, which is an integral part of the dermis and blood vessel wall, is degraded by the activity of the enzyme hyaluronidase, which can be inhibited by the protective roles of flavonoids [3]. Additionally, phenolic compounds may have antimicrobial properties and assist in the preservation of cosmetic products against secondary infections [3].

Low drug permeability through human epidermis can be ameliorated using penetration enhancers (compounds such as surfactants, terpenes, lipophilic solvent, and fatty acids), which modify the permeability of the skin barrier reversibly. Rutin, catechin, epicatechin, and quercetin have a limited penetration. Quercetin shows poor permeability even in the presence of the enhancers, due to its absolute insolubility in water [35]. Several authors suggested that the permeability of this compound can be increased by uploading flavonoids in liposomes or other kinds of carrier systems. It can also be affected by the presence of promoters of permeability, such as glycerin and propylene glycol [3]. In the skin, HSD may significantly stimulate epidermal hyperplasia and improve epidermal permeability through epidermal proliferation, osteoblasts differentiation, and lipid secretion. Different skin layers and skin proteins can respond differently to HSD treatment. Involucrin, present in the stratum spinosum, does not respond to HSD treatment. Filaggrin and loricrin (located in the stratum granulosum and stratum spinosum) expressions are increased. These beneficial effects that improve epidermal permeability and regulate filaggrin can be useful in the treatment of certain skin disorders, such as cutaneous inflammation and atopic dermatitis [52]. In vitro, HSD can inhibit the tyrosinase in melanocytes and reduce the process of melanogenesis, or influence melanocytes proliferation. Therefore, the possibility of affecting tyrosinase activity, and the practical application thereof in the field of anti-ageing preparations designed to prevent lentigo senilis and lentigo solaris makes HSD remarkably interesting for skin whitening [3]. Daily topical applications of HSD microemulsion have shown a significant skin whitening effect, reduction in trans-epidermal water loss, and inhibition of irritation effect after exposure to UV rays after four weeks. In their study, Kim et al. [56] showed that HSD had a depigmentation effect by blocking the melanophilin, a tripartite protein complex which is a response for transport of the melanosome into the melanocytes. This was proven by melanosome aggregation tests in cells. HSD did not inhibit melanin production in melanoma cells, but reduced skin pigmentation in the reconstruction of human epidermal skin [56]. HSD may also affect polyoxygenase, cyclooxygenase, hyaluronidase, collagenase, elastase, and tyrosinase; thus, it can contribute to the reduction in modifications in the skin’s connective tissue remodeling [3]. Hyaluronidase plays a significant role in regulating the permeability of capillary walls and supporting tissues by causing the breakdown of hyaluronic acid and increasing the permeability of the tissue [5]. The HSD’s ability to chelate multivalent metals such as iron(III), iron(II), copper(II), zinc(II), and manganese(II) is as relevant to the inhibition of enzymes, which contain metal ions in their reaction center or require metal cations cofactors, as to the inhibition of inflammatory processes and function of vascular vessels [3,15,57,58]. Chelation properties of HSD are particularly important for skin bleaching activities, as well as HSD interaction with the tyrosinase during catalytic production of melanin. Tyrosinase has copper(II) in the active site, which is responsible for tyrosinase oxidation activities [9].

## 4. Materials and Methods

### 4.1. Materials

Oranges (*Citrus sinensis*) and orange peel bagasse were purchased from a local supermarket in Campinas (Sao Paulo, Brazil). Hesperidin standard was bought from Sigma-Aldrich (St. Louis, MA, USA) at analytical grade (>97%). Calcium(II) chloride, copper(II) acetate, sodium hydroxide, and hydrochloric acid were from Labsynth (Diadema, Sao Paulo, Brazil) at practical grade. Deuterated solvent (DMSO-*d*_6_) was purchased from Sigma-Aldrich (St. Louis, MA, USA). Methanol and phosphoric acid for the UHPLC assays were analytical grade (Avantor “Performance Materials”, Mexico^®^, Ecatepec de Morelos, Mexico). ABIL^®^ Care 85 (Bis-PEG/PPg-16/16 PEG/PPG16/16 Dimethicone; Caprylic/Capric Triglyceride) were sampled from the EVONIK^®^ group (Essen, Germany). Muru muru (*Astrocaryum murumuru* Seed Butter), Cupuaçu butter (*Theobroma grandiflorum* Seed Butter), and Andiroba oil (*Carapa Guianensis* Seed Oil) were sampled from Amazon oil industry; Glyceryl monostearate, Mineral oil, Carbopol ULTREZ 10, Triethanolamine, and Vit E-acetate (α-Tocopheryl acetate) were bought from Fagron^®^ (Rotterdam, The Netherlands).

### 4.2. Extraction of HSD from Humid Orange Bagasse

The fresh chopped orange peel (*Citrus sinensis,* 100 g) was milled in a blender and immersed in 300 mL of aqueous CaCl_2_ solution (7.5%). The mixture was left to stand for 24 h, then, a NaOH 1 mol L^−1^ solution was added dropwise until a pH of 11.50 was reached. It was set to stand for 24 h, the pH was measured, and it dropped to 9.66 overnight. Then, the suspension was filtered, and the pellet was much harder when compared with the starting material, probably because of the interactions between calcium(II) ions and pectin. Concentrated HCl was then added to the solution, and pH was lowered to 4.35. The solution was left to stand at 4 °C for 2 days. After 24 h, crystals were formed at the bottom of the glass. However, for the best yields, two days of precipitation is recommended. After 48 h, HSD was filtered onto a MILIPORE^®^ (Burlington, MA, USA) apparatus, using a 0.45 μm pore filter, and dried for 2 h in drying oven (60 °C).

### 4.3. Nuclear Magnetic Resonance—NMR

Nuclear Magnetic Resonance spectra were acquired on a Bruker *AVANCE III* 600 MHz spectrometer (Bruker Biospin, Karlsruhe, Germany) equipped with a 5 mm Triple Resonance Broadband Inverse (TBI) probe at 25 °C. ^1^H NMR (1D, 600.173 MHz) spectra were acquired using a f1 pre-saturation sequence with 32 transient scanning, 64 k data points, and 13 kHz bandwidth. ^13^C NMR spectra were obtained using the 1D sequence with power decoupling. Isolated HSD was dissolved in DMSO-*d_6_* (20 mg mL^−1^). Two-dimensional spectra (Heteronuclear Single Quantum Coherence—HSQC) were acquired using the 600.17 MHz frequency domain spectrometer F2 and 150.91 MHz in the F1 domain with a free induction decay size (FID) of 4096 (F2) and 256 (F1) data points and 16 “ghost” scans were used. HSQC (*hsqcedetgpsp.*) spectra were recorded with an acquisition time of 2.129 × 10^−1^ s (F2) and 4.437 × 10^−3^ s (F1) pulse sequence. The Heteronuclear Multiple Bond Correlations (HMBC) were acquired using the 600.17 MHz frequency domain spectrometer F2 and 150.91 MHz in the F1 domain with a free induction decay size (FID) of 2048 (F2) and 256 (F1) data points and 16 “ghost” scans were used. HMBC spectra were acquired with an acquisition time of 2.130 × 10^−1^ s (F2) and 4.437 × 10^−3^ s (F1), using the *hmbcgplpndgf*. pulse sequence. The spectra were processed in MestreNova software, using LB = 1.00 for F2 and LB = 0.30 for F1.

### 4.4. Fourier Transform Infrared Spectroscopy—FT-IR

The Fourier transform infrared spectrum (FT-IR) of HSD was obtained on a Cary 630 FT-IR spectrometer equipped with a monolithic diamond attenuated total reflection (ATR) accessory (Agilent Technologies Inc., Santa Clara, CA, USA). The spectra were recorded accumulating 128 scans at a resolution of 4 cm^−1^ in the range from 4000 to 400 cm^−1^.

### 4.5. UHPLC—Purity of Extracted Hesperidin

The HSD purity was determined utilizing Ultra-High-Performance Liquid (UHPLC—Waters, Milford, MA, USA) with the reverse stationary phase (Zorbax Eclipse C18, length 10 cm, 5 μm, Agilent Technologies Inc., Santa Clara, CA, USA). The mobile phase was prepared with methanol and water (60:40 *v*/*v*) with the addition of phosphoric acid (0.1%). A hesperidin analytical standard (Sigma-Aldrich, >97% of purity, St. Louis, MO, USA) was used for calibration curve construction from the injection of aliquots of 1, 2, 4, 6, 8, and 10 μL of standard solution, HSD in methanol (100 ppm). Calibration curves and the analysis of the sample from the extraction were performed in triplicate. The following method was adapted from Weon et al. (2012) [59].

### 4.6. Chelation Properties of Hesperidin

Chelation properties of HSD were tested through the interaction of HSD with copper(II), following the procedure in Selvaraj et al. [58]. For the experiment, HSD (1 mmol L^−1^) was dissolved in 50 mL of methanol, mixed with 0.199 g of copper(II) acetate in 25 mL of double distilled water and stirred for 6 h at room temperature. The pale green insoluble precipitate was obtained and filtered and washed with water and methanol to remove the excess of HSD and copper(II) acetate. The product was then air-dried. The greenish precipitate obtained was vacuum dried and characterized.

### 4.7. Fluorescence Suppression—Quenching

The measurements were performed on a Varian spectrofluorometer (Cary Eclipse model, Agilent Technologies Inc., Santa Clara, CA, USA) by applying a wavelength of 278 nm for excitation of the collagenase, excitation and emission slits were 5 nm wide, and acquisition of emission spectra occurred in the range of 300–450 nm, with the sample compartment with a thermostat at 25, 30, and 37 °C. A 10 × 10 mm optical path quartz (Hellma, Mülheim, Germany) cuvette was used. To obtain spectra, 2 mL of collagenase (Sigma-Aldrich) in concentration 0.065 mg mL^−1^ (water solution) were titrated with hesperidin solution in DMSO (0.10 mg mL^−1^ concentration). Additions of aliquots of HSD were as follow: 0.5 μL up to 4 μL, 1.0 μL up to 50 μL, and 5 μL up to a volume of 110 μL, reaching the final concentrations of 5.68 mg mL^−1^ of HSD in solution.

### 4.8. Preparation of Hesperidin Nanocrystals

HSD was utilized to formulate nanoemulsions. Polymers utilized in preparations were poloxamer (Pluronic F127, Sigma Aldrich, St. Louis, MO, USA) and nanocellulose, which served as stabilizers and showed different capacities regarding the preservation of hesperidin nanostructure. These were dissolved in water and mixed with an Ultra Turrax mixer for 15 min at a speed of 14,000 rpm min^−1^. Pre-milling with the mixer avoided blocking the homogenizing gap of the homogenizer present in the prepared formulation. Hereupon, the formulations I, II, and VII were processed through the homogenizer (Gea Niro Soavi, Model NS 1001L—Panda 2k, Dusseldorf, Germany) at high pressure (600 bar) for 5 cycles. During the homogenization process, a pressure decrease was observed, especially after the third cycle. The process utilized for HSD nanonization for formulations III, IV, V, and VI was based on the technology NANOEDGE™, where HSD was solubilized in a minimum quantity of solvent (or homogenous mixture of complex composition). The resulting solution was added to a mixture of purified water and stabilizers and it underwent a homogenization process with high-potency ultrasound (Ultronique, Model: DESRUPTOR; Freq. US: 20 kHz; Potency US: 750 W, Indaiatuba SP, Brazil), 70% potency for 30 min. Formulations V and VI were prepared through a NANOEDGE-like technique, where initially HSD was dissolved in the mixture of essential oil (orange peels) and glycerol and added into the solution of purified water and nanocellulose; afterwards, it was sonicated. Formulations prepared using two nano-techniques and their ingredients were as follows:

I—H69, combination of pre-milling with an Ultra Turrax and high-pressure homogenization, ingredients used: hesperidin and water;

II—H69, combination of pre-milling in an Ultra Turrax and high-pressure homogenization, ingredients used: hesperidin, DMSO, glycerol, sodium-carboxyl methyl cellulose, and water;

III—NANOEDGE-like, combination of microprecipitation and ultrasound, used ingredients: hesperidin, DMSO, glycerol, poloxamer (Pluronic F127), and water;

IV—NANOEDGE-like, combination of microprecipitation and ultrasound, used ingredients: hesperidin, DMSO, orange oil, poloxamer (Pluronic F127), and water;

V—NANOEDGE-like, combination of microprecipitation and ultrasound, used ingredients: hesperidin, propylene glycol, nanocellulose, and water;

VI—NANOEDGE-like, combination of microprecipitation and ultrasound, used ingredients: hesperidin, glycerol, nanocellulose, and water;

VII—H69, combination of pre-milling in an Ultra Turrax and high-pressure homogenization: hesperidin, glycerol, orange oil, poloxamer (Pluronic F127), and water;

VIII—Milling in an Ultra Turrax, ingredients used: hesperidin, glycerol, orange oil, poloxamer (Pluronic F127), and water.

### 4.9. Formulation of the Silky Cream

Oil-in-water silky cream formulations were prepared following the protocols of the EVONIK^®^ group (Essen, Germany), for use of the surfactant ABIL^®^ Care 85 (Bis-PEG/PPg-16/16 PEG/PPG16/16 Dimethicone; Caprylic/Capric Triglyceride) and presented in Table 3. Muru muru (*Astrocaryum murumuru* Seed Butter), Cupuaçu butter (*Theobroma grandiflorum* Seed Butter), and Andiroba oil (*Carapa Guianensis* Seed Oil) were sampled from Amazon oil industry; Glyceryl monostearate, Mineral oil, Carbopol ULTREZ 10, Triethanolamine, and Vit E-acetate (α-Tocopheryl acetate) were from Fagron^®^. Emulsifier ABIL^®^ Care 85 that gives a velvety-silky skin feel was sampled from EVONIK^®^ and was used in exceptionally low usage concentration. Ethylhexylglycerin and Phenoxyethanol (Fagron^®^) were used for formulation preserving in exceptionally low concentration, instead of the commercial paraben’s components.

Ingredients of the phases A and B were measured, transferred into two beakers, and heated to 50 °C. When the oil phase was homogenized and all the ingredients were molten, phase A was added into phase B. The mixture was stirred and homogenized until it cooled down to a temperature of 25 °C. Ingredients of the phases C and E were added at room temperature, along with stirring. A whitish cream, oil-in-water, silky, with soft consistency, and easy to apply with no greasy film leftovers after applying was obtained. Nanoemulsion VI was used for the preparation in the A2 cream and Nanoemulsion VII was used for preparation in the A3 cream formulation.

### 4.10. Skin Model USP-FTS Skin Corrosion Test of Cream Formulations

Full thickness skin models (FTS) for the toxicity test of hesperidin cream formulations were prepared in the Laboratory of Biology of the Skin, coordinated by Prof. Silvya Stuchi Maria-Engler, Faculty of Pharmaceutical Sciences, University of Sao Paulo. Fibroblast and keratinocytes were isolated from the normal human skin cells, from donated foreskin samples obtained from the University of Sao Paulo Hospital (Sao Paulo, Brazil). Performed assays were under the approval of the local Ethics Committee (HU CEP Case No. 943/09 and CEP FCF/USP 534). Isolated cells were seeded on top of the collagen I matrix model in a 6-well plate containing enough specific medium mixture for the FTS model as to maintain the skin at the air–liquid interface, as it is described by Catarino et al. [32]. For the corrosion tests, the following system was used: 100 µg mL^−1^ of hesperidin in 10% DMSO, cream without the hesperidin nanoemulsion (A1), two formulations of hesperidin nanoemulsions (Nanoemulsion VI and Nanoemulsion VII), and the last two in the cream formulations as A2 and A3 creams. As a negative control, 0.9% sodium-chloride solution was used. The 100 µg mL^−1^ of hesperidin in 10% DMSO were tested by adding 5 mL in the well plate, and the 0.5 mL of cream formulations were applied over epidermis. Samples were left over night for incubation (37 °C, 5% CO_2_). All tests were performed in duplicate. After 24 h, skins were washed with 0.9% of sodium-chloride solution and preserved in paraffin for further microscopic tests.

### 4.11. Diffraction Light Scattering and Zeta Potential

Mean size, polydispersity, and zeta potential of nanoparticles from each formulation were determined using diffraction light scattering technique (Nano ZS Zetasizer—Malvern, PANalytical Almelo, The Netherlands). Known also as Photon Correlation Spectroscopy—PCS, this is the technique that was used to measure the hydrodynamic diameter of particles in a range from microns up to 1 nm.

### 4.12. Nanoparticle Tracking Analysis—NTA Analysis

Analysis (NTA) was performed with a NanoSight LM20 microscope (NanoSight, Amesbury, UK), equipped with a 640 nm laser sample camera and a fluoroelastometer. Samples were diluted and injected with sterile syringes 57 (BD Discardit II, Jersey, USA) until the liquid reached the tip of the mouthpiece. All measurements were performed at room temperature, with live monitoring of thermal stresses. The software used to capture and analyze the data was NTA 2.0 Build 127. The samples were captured with 60 s time using the parameters predetermined by the manual.

### 4.13. Electronic Microscopy—Transmission, Scanning, and Atomic Force

Utilizing the microscope SEM JEOL JSM-6360 LV (JEOL, Akishima, Tokyo, Japan) and an atomic force microscope (Shimadzu, SPM 9500J3), the size of the nanoparticles in the nanoemulsions was verified. The sonicated hesperidin nanoemulsion was dropped onto a sample holder, dried at room temperature, and coated with gold or platinum using a MED 020 Sputter (BalTech, Balzers, Liechtenstein). The images were obtained using an acceleration voltage of 10 kV and a secondary electron detector.

## 5. Conclusions

Hesperidin (HSD) was obtained from citrus peels through a pH-triggered precipitation method conducted just in water. The extracted hesperidin purity and yield were excellent when compared to standard methods of its extraction and a commercially available compound. HSD showed chelating activity toward bivalent ions and interacted with collagenase. HSD was used for the formulation of a promising anti-ageing face cream. Upon nanonizing, HSD nanoparticles (150 to 400 nm) were employed into up to 12 months stable nanoemulsions, whose applicability was tested in vitro on artificial skin. No nocive effects were observed, and the best face cream formulation (cream A2) showed good results in reducing the black circles in the under eye region. Moreover, daily topical applications of hesperidin nanoemulsion have shown a significant skin whitening effect, reduction in trans-epidermal water loss, and inhibition of an irritation effect after exposure to UV rays. This way, a clean cream active ingredient is proposed, and a promising cosmeceutical was formulated starting from agro-industrial waste.

## Figures and Tables

**Figure 1 molecules-25-04534-f001:**
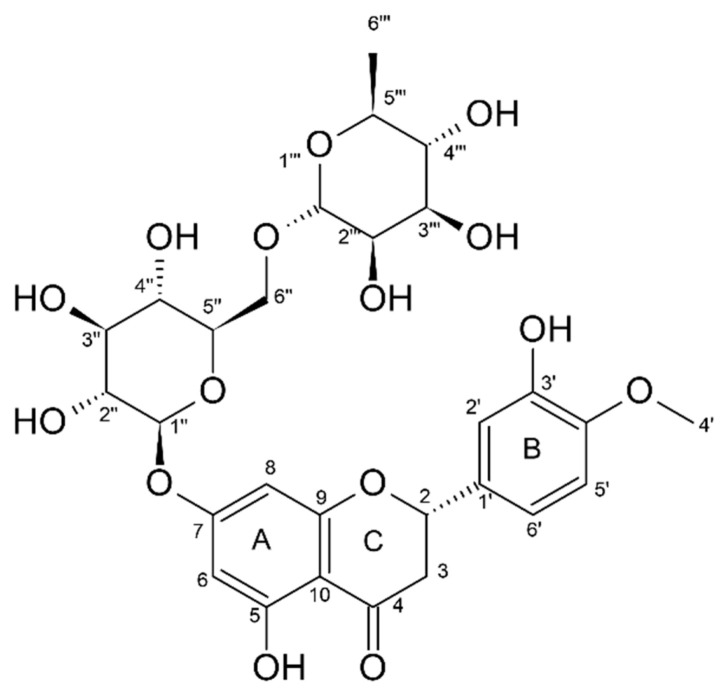
Chemical structure of hesperidin (2*S*)-5-hydroxy-2-(3-hydroxy-4-methoxyphenyl)-7-[(2*S*,3*R*,4*S*,5*S*,6*R*)-3,4,5-trihydroxy-6-[[(2*R*,3*R*,4*R*,5*R*,6*S*)-3,4,5-trihydroxy-6-methyloxan-2-yl]oxymethyl]oxan-2-yl]oxy-2,3-dihydrochromen-4-one). HSD carbon atoms are numbered, and A–C rings are shown.

**Figure 2 molecules-25-04534-f002:**
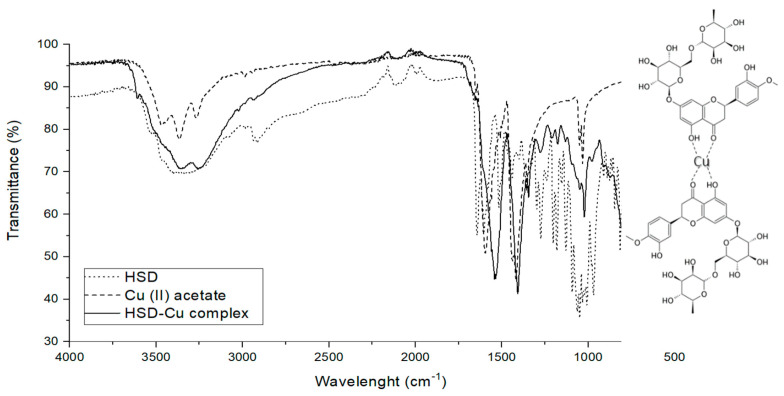
Fourier transform infrared (FT-IR) spectra of hesperidin (HSD, dotted line), copper(II) acetate (Cu(II), dashed line), and HSD–Cu(II) complex (full line) on a Cary 630 FT-IR spectrometer equipped with a monolithic design diamond Attenuated Total Reflectance (ATR) accessory (Agilent Technologies Inc.). Hesperidin is a chelating agent of metal multivalent ions (Me^2+^). The creation of the complex involves -OH groups in ortho- positions that coordinate with divalent cations [1].

**Figure 3 molecules-25-04534-f003:**
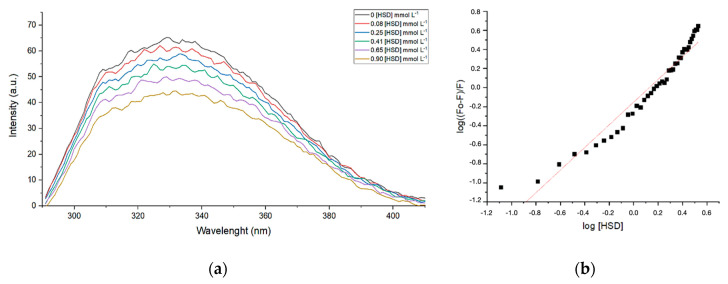
Illustration of fluorescence spectroscopy data: (**a**) Collagenase (*Clostridium hystolicum*) at 37 °C upon addition of HSD (in DMSO) at concentrations from 0.08 to 0.90 mmol L^−1^. (**b**) Diagram of log((F_0_−F)/F) in function of log[HSD].

**Figure 4 molecules-25-04534-f004:**
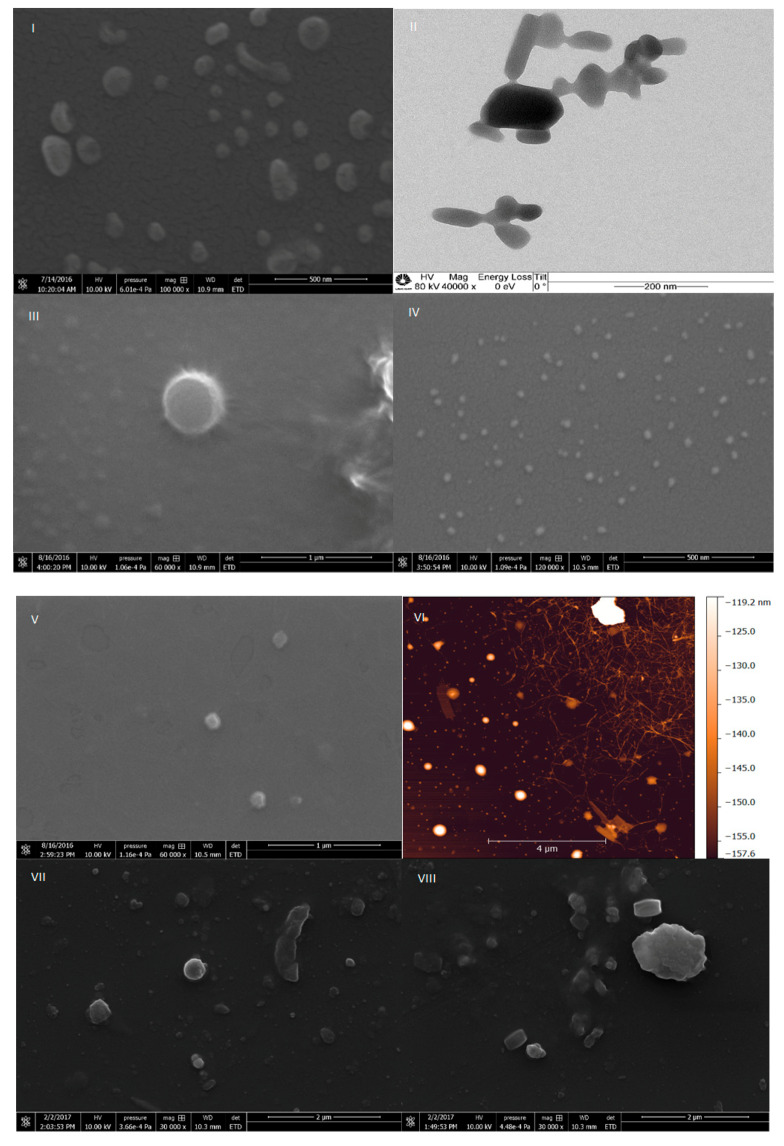
Scanning electron micrographs (**left**; scale bar = 500 nm) of formulation **I** and transmission electron micrographs (**right**; scale bar = 200 nm) of formulation **I**. Scanning electron micrographs of formulation **III** (**left**; scale bar = 1 µm) and formulation **IV** (**right**; scale bar = 500 nm), respectively. Scanning electron micrographs of formulation **V** (**left**; scale bar = 1 µm) and atomic force micrographs of formulation **VI** (**right**; scale bar = 4 µm). Scanning electron micrographs of formulations **VII** and **VIII** (**left**; scale bar = 2 µm).

**Figure 5 molecules-25-04534-f005:**
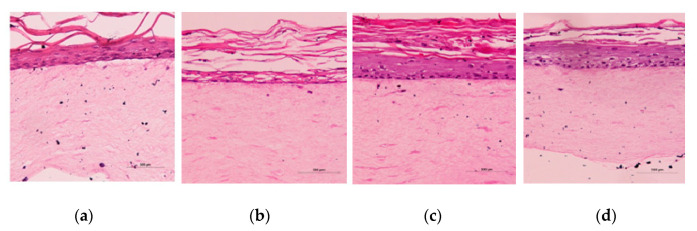
Photomicrographs of Histological Analysis—Eosin stain (sample in paraffin) of the full thickness skin (FTS) equivalent skin model after the cream application assay (**a**) negative control: sodium chloride 0.9%; (**b**) positive control: HSD-DMSO, (**c**) A1 cream, and (**d**) A2 cream. The skin models were generated using primary human keratinocytes and fibroblasts—the FTS model; (Scale bar = 500 µm).

**Table 1 molecules-25-04534-t001:** Thermodynamic parameters of collagenase–HSD interaction.

	Temperature °C	ΔG (kJ mol^−1^)	ΔH (kJ mol^−1^)	ΔS (J mol^−1^ K^−1^)
**Collagenase–Hesperidin**	25	20.0		
	30	22.2	−14.5	−19.0
	37	22.6		

**Table 2 molecules-25-04534-t002:** Mean size, zeta potential, and polydispersity (PdI) of nanoparticles from different formulations prepared with various techniques, measured by a Malvern Zetasizer and a NanoSight LM20 microscope.

Formulation	Dynamic Light Scattering (DLS)			Nanoparticle Tracking Analysis (NTA)		
	Mean Size [nm]	Zeta Potential [mV]	PdI	Mean [nm]	Standard Deviation [nm]	Conc. of Particles [E8/mL]
I	148.0	−36.2	0.181	109.0	39.0	2.03
II	1126	−29.6	0.579	157.0	59.0	0.14
III	265.7	−4.81	0.332	226.0	55.0	0.63
IV	259.7	−32.7	0.370	73.0	15.0	0.27
V	335.6	−30.1	0.375	63.0	18.0	0.43
VI	367.2	−33.6	0.332	111.0	47.0	5.65
VII	287.7	−19.0	0.433	76.0	15.0	2.34
VIII	485.4	−51.6	0.506	33.0	4.00	2.48

**Table 3 molecules-25-04534-t003:** Silky velvet cream formulations prepared for application of hesperidin nanoemulsions.

Phase	Name	A1 (%, g g^−1^)	A2 (%, g g^−1^)	A3 (%, g g^−1^)
A	Bis-PEG/PPG-16/16 PEG/PPG-16/16 Dimethicone; Caprylic/Capric Triglyceride	1.0	1.0	1.0
	Muru muru butter	1.0	1.0	1.0
	Cupuaçu butter	1.0	1.0	1.0
	Andiroba oil	3.0	3.0	3.0
	GMS	2.0	2.0	2.0
	Cetostearyl alcohol	3.0	3.0	3.0
	Mineral oil	5.0	5.0	5.0
B	Carbopol ULTREZ 10	0.2	0.2	0.2
	Aqua (Purified Water)	79.8	-	-
	Nanoemulsion	-	79.8	79.8
C	Triethanolamine	5 *gtts	5 *ggts	5 *ggts
B	Vit E-acetate	0.5	0.5	0.5
E	Ethylhexylglicerin, Phenoxyethanol	0.5	0.5	0.5

*gtts is drops.

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
