# Peer review of "New Sustainable Process for Hesperidin Isolation and Anti-Ageing Effects of Hesperidin Nanocrystals"

_molecules, 2020, doi:10.3390/molecules25194534_

Round 1
Reviewer 1 Report
In the submitted manuscript Stanisic et. al. proposed a green method for extraction of hesperidin from orange peels, which does not involve the use of organic solvents. Furthermore, the methods for hesperidin nanonization were explored, after which the nanoemulsions were incorporated into cream formulations that were tested in vitro on the artificial 3D skin.
Although this work is interesting, there are a few comments that should be addressed before its publication:
The statement in lines 16-18 is not clearly written.
Line 48: Very recently it was also revealed that natural polyphenolic compounds, such as 6-gingerol, possess promising cancer-preventing potential against various chemical carcinogens.
Reference: Int. J. Mol. Sci. (2020) 21, 695
In lines 153-154 it is written: »Hesperidin chelates zinc(II) ion, which is in the catalytic site of the enzyme (collagenase), and changes the conformation of the protein.« To support those claims, the authors should conduct additional chelation experiments with hesperidin and Zn2+ ions as they did with Cu2+ ions.
Line 156: The references are missing.
Lines 156-158: How were the thermodynamic values of enthalpy (ΔH) and entropy (ΔS) of collagenase-hesperidin interaction determined?
The fluorescence spectroscopy data from Figure 3 should be discussed in more detail. The results at 25 ⁰C and 30 ⁰C should also be added and evaluated.
Lines 174-176: The value of ∆H, not ∆G is negative. The obtained thermodynamic data from Table 1 should be critically evaluated and discussed in more detail.
Lines 188-189: What was the reason for such variations in zeta potentials (Table 2)?
Lines 210-219: What about nanoemulsions VII and VIII? Which nanoemulsion was the most stable?
Line 269: The authors should comment on which formulation has the biggest potential on normal skin.
Line 299: Beneficial effects of various flavonoids have also been reviewed in the recent articles: Molecules (2016) 21, 901 AND Nutrients (2019) 11, 257
Line 300: Very recently new inverse molecular docking protocol was proposed to identify potential human protein targets of polyphenol curcumin, which could provide further insights into molecular mechanisms of polyphenolic compounds from various natural sources.
Reference: Molecules (2018) 23 (12), 3351
Lines 472-474: It is not clearly written, which nanoemulsions were used in formulations A2 and A3.
Lines 505-508: It should be written, which face cream formulation was the best. Moreover, further conclusions are speculations as they were not supported by experiments conducted in this work.
Author Response
Thank you for the opportunity to improve our manuscript, critical review, and very constructive suggestions. We have revised our manuscript following the questions raised and point-to-point answers are attached to this message.

Reviewer 2 Report
ABSTRACT
Rewrite with the use of shorter sentencies.
INTRO
Fig.1
1
Use better reproduction (=>dpi) or draw a new graphic file using any common chemical software.
2
If you mean to show HSD, be careful with chirality of C-2. Use proper line instead of "any", according to the presented chemical name.
3
This work is not a chemical study. To be more reader-friendly, you are allowed to use common semi-systematic name like 'alpha-L-rhamnopyranosyl(1'''->2'')beta-D-glucopyranosyl(1''->7)...' instead of IUPAC '...trihydroxy-6-methyloxan-2-39 yl]oxymethyl]oxan-...'.
l.75 'cathechin'=>'catechin'
l.316 'cyaniding'=>'cyanidin'
l.88-90
The use of organic solvents is eco-unfavourable but, on the other hand, fasten the isolation. That is sometimes necessary to avoid fermentation process during isolation) => Refer to this aspect in Discussion, use appropriate Refs if needed.
RESULTS / DISCUSSION
Chapter 2.1
In Discussion, compare your yield and purity with standard methods. Use Refs.
Is the 7.5% CaCl2 universal or it depends on the amount of pectins that can be different in each batch of orange peels. Refer to this problem in Discussion.
Fig.S1
Verify placing of letters in figure, add (e).
BTW, why not to use centrifugation and washing the precipitate with appropriate solvent instead of filtration. It is usually more effective. Particularly in industrial scale.
Chapter 2.2 & Fig.S2
HSQC clearly shows the impurity (about 6.5/115ppm). If 600 MHz NMR was applied, why not to run qHNMR and calculate the purity in a modern way, using an internal standard of the highest purity. Instead of peak areas calculation on chromatogram (in 280 nm). Especially, because te shape of presented peak is poor and its Rt is low when compared with its intensity, as shown in Fig.S4. It may lead to omission of impurities and improper calculations. Refer to this.
Without validated assay you can name it 'approximate purity'.
Your purity is higher than of Aldrich standard. Refer to this.
BTW, what about the chiral purity?
What is the source of HPLC method? Add reference here or in Materials &Methods, or declare that it is yours.
Fig.S3
Presented HMBC has extreme tailing in y (delta C) scale.
Fig.2
Be consequent, 'Hsd'=>'HSD'.
Use em-dashes consequently in all work, e.g. l.150;
similarly, en-dashes without spaces in situations like in l.288 (0.5-0.7)
Chapter 2.3 & Fig.3
Is [Q] equal to [HSD]? If so => unify.
Chapetr 2.4
The 12M-stability of formulations was analyzed after storage in 4°C but, in Abstract, a reference to tropical-zone formulation is given. Refer to this.
MATERIALS & METHODS
Define the origin and classes of chemicals. Define the origin of plant material, as well as the species (orange is ambiguous; sweet, bitter), and stage of fruits maturity.
l.364
Was HSD crystalline or amorphic? How it was judged? Refer to this.
l.365 'in'=>'is'
l.367 use degree sign
l. 373
Place the concentration at the end of sentence (after DMSO).
l. 449
Non-standard names should be explained with full botanical names and plant parts. (Muru muru, Cupuacu, Andiroba).
ch.4.9.
Is skin donation a topic for a Bioethical Comission? Refer to this problem here.
l.528
Add the numbers of received grants
l.667
Define the availability of samples and formulations.
Author Response
Thank you very much for the opportunity to revise and improve our manuscript. We have found all suggestion and critical comments very useful. Please find inclosed the rebuttal letter with point-to-point answers.

Round 2
Reviewer 1 Report
The authors successfully resolved all issues raised by this reviewer. Consequently, the manuscript has been significantly improved and can be in its current version recommended for publication in Molecules.
Reviewer 2 Report
Dear Authors,
1/
In text, declare that you received the assay approval from suitable Bioethical Comission. Add its name and number of document.
2/
English needs to be checked.
Good luck.